# Endometriosis and In Vitro Fertilization

**DOI:** 10.3390/medicina60081358

**Published:** 2024-08-21

**Authors:** Sania Latif, Shirin Khanjani, Ertan Saridogan

**Affiliations:** 1Reproductive Medicine Unit, University College London Hospital, London NW1 2BU, UK; 2EGA Institute from Women’s Health, University College London, Londen WC1E 6AU, UK

**Keywords:** endometriosis, infertility, in vitro fertilization

## Abstract

In vitro fertilization (IVF) is an established option for the management of infertility in patients with endometriosis, though there remains ongoing debate around the extent to which endometriosis may compromise IVF treatment success, in which fertilization and preimplantation embryo development occur outside the pelvis. Whether endometriosis impacts oocyte and embryo quality and/or endometrial receptivity remains central to this debate. Here, we review the current literature relating to the impact of endometriosis on IVF outcomes and management strategies to consider when performing IVF treatment.

## 1. Introduction

Endometriosis is commonly found in women who experience conception delay, and women who are diagnosed with endometriosis are nearly twice as likely to suffer from infertility [1]. Endometriosis has been proposed to adversely affect chances of natural conception due to mechanical distortion of pelvic anatomy, impairment of gamete and embryo transport, reduced oocyte quality and embryo quality, diminished ovarian reserve, altered immune and endocrine function, dysregulation of hormonal and cell-mediated functions involved in endometrial receptivity, and inability to have regular intercourse [2,3,4,5,6,7,8,9].

In vitro fertilization (IVF) is an established treatment option for the management of infertility in patients with endometriosis [10,11]. During IVF treatment, oocytes and sperm are brought together outside the body (in vitro), which has been proposed to mitigate the toxic effects of the endometriotic pelvic environment. We review the current literature regarding the impact of endometriosis on IVF treatment results and management strategies to consider when performing IVF treatment.

## 2. Methods

A PUBMED search was performed for articles published between October 1991 and May 2024. We used the MeSH search terms: ‘endometriosis’, ‘in vitro fertilization’ OR ‘assisted’ AND ‘fertilization’. Restrictions to the English language and human species were applied. We included clinical trials, systematic reviews, and meta-analyses and screened the references of the selected articles to identify studies of relevance. SL and ES independently extracted data.

## 3. Endometriosis and IVF Outcomes

In an early meta-analysis, it was reported that women with ASRM stage I/II endometriosis with tubal factor infertility have lower fertilization and implantation rates compared to women with tubal factor infertility without endometriosis [12]. In addition, it was found that both oocyte yield and fertilization rate were affected by endometriosis of any stage. The authors concluded that endometriosis can negatively impact oocyte and embryo quality, and endometrial receptivity [12]. Notably, this meta-analysis included older studies that were published prior to 2000 when ART success rates were substantially lower compared to more recent success rates.

A subsequent meta-analysis published in 2013 reported lower fertilization rates but no significant difference in implantation, clinical pregnancy, or live birth rates in women with ASRM stage I/II endometriosis compared to women without endometriosis, whereas women with stage III/IV endometriosis had reduced implantation and clinical pregnancy rates but no difference was found in live birth rates [13]. Hamdan and colleagues found that women with stage III/IV disease had significantly lower clinical pregnancy rates and live birth rates with a lower number of oocytes retrieved when compared to controls [14].

An analysis of a large body of data from the Latin American Data Registry has since shown that although women with endometriosis undergoing IVF had a lower oocyte yield and higher cycle cancellation rates, these findings did not translate to a difference in live birth rates [15]. Similar findings have been reproduced in several other meta-analyses [16,17,18,19].

### 3.1. Oocyte/Embryo Quality

Successful IVF treatment is dependent upon oocyte quality, which in turn impacts the creation of top-quality embryos, and subsequently, a receptive endometrium is crucial for implantation of the embryo. In 2022, a large study scrutinized 13,614 IVF cycles performed in women with endometriosis undergoing either autologous or donor oocytes, using data from the Human Fertilization and Embryology Authority (HFEA). They reported no significant difference in live birth rates in either fresh or frozen embryo transfer cycles, suggesting minimal or no impact of oocyte quality in patients with endometriosis [20]). Moreover, studies examining euploidy and aneuploidy rates in women with endometriosis compared to age-matched controls have reported equivalent euploidy rates between groups [21,22]. A list of studies that have evaluated the effect of endometriosis on oocyte and embryo quality are presented in Table 1. Overall, although presence of endometriosis is considered to impact oocyte quality and in return embryo quality, a clear clinical impact on IVF outcomes is not confirmed.

### 3.2. Endometrial Receptivity

Endometrial receptivity is dependent upon an appropriate hormonal exposure of estrogen and progesterone and the regulated expression of numerous genes [36]. The eutopic endometrium of women with endometriosis has been shown to exhibit characteristics of progesterone resistance and aberrant cell signalling, suggested to alter endometrial receptivity [36].

However, in a study investigating transcriptomic modifications using the endometrial receptivity array (ERA) test, there was no difference in the endometrial receptivity gene signature during the implantation window in women with endometriosis compared to healthy controls [37]. A sibling oocyte study evaluating the impact of surgically diagnosed stage III-IV endometriosis on IVF outcomes where women had received oocytes from the same donor, also found no difference in implantation rates in women with severe endometriosis (*n* = 25) compared to women without endometriosis (*n* = 33), suggesting no negative impact of endometriosis on implantation rates in women undergoing IVF ([4]. A randomized control trial investigating the effectiveness of endometrial scratching prior to IVF has shown that there was no significant difference in implantation rate or live birth rate in women with endometriosis compared to healthy controls, confirming that there is no role for performing endometrial scratching to improve implantation rates [38].

Most recently, a landmark retrospective cohort study including 459 euploid frozen embryo transfer cycles comparing women with surgically diagnosed endometriosis to women undergoing IVF for isolated male factor infertility showed that there was no difference in implantation rates, clinical pregnancy, pregnancy loss, or live-birth rates [22]. They reported that blastocyst development, number of good-quality blastocysts produced, and euploidy rates were the same in both groups [22].

The current evidence suggests that when known high-quality embryos are transferred during a medicated frozen embryo transfer cycle, women with endometriosis do not exhibit any marked defect in endometrial receptivity [22].

### 3.3. Ovarian Endometriomas

Endometriomas are prevalent in almost 50% of women with endometriosis and reduce ovarian reserve due to damage to healthy ovarian tissue and mechanical stretch, resulting in a progressive reduction in the pool of primordial follicles [39]. Meta-analyses examining the impact of ovarian endometriomas on IVF/ICSI outcomes have shown a significant reduction in the number of mature oocytes retrieved in women with endometrioma versus controls but no difference in the gonadotropin dose and duration, the total number of embryos, high-quality embryos, clinical pregnancy rate, and live birth rate between women with and without ovarian endometriomas [14,16,17]. The reduction in the number of oocytes retrieved is greater in women with large endometriomas and where endometriomas are present bilaterally [40].

Surgery can adversely impact ovarian reserve due to the inadvertent removal of healthy ovarian tissue during cystectomy, thermal damage to the ovarian cortex during haemostasis or ablation, vascular compromise, and local inflammation. After surgical removal, there is a reduction in anti-Mullerian hormone (AMH) levels in unilateral endometriomas by 30% and in bilateral endometriomas by 44% [41], with a reported risk of premature ovarian insufficiency of 2.4% following bilateral ovarian endometrioma removal [42]. Younger women have a higher recurrence rate of endometriomas requiring repeated surgery, compounding the insult to their ovarian reserve [43]. A working group of the World Endometriosis Society, European Society for Gynaecological Endoscopy and European Society for Human Reproduction and Embryology has collaborated in developing recommendations on the practical aspects of endometrioma surgery to reduce their adverse impact [44].

## 4. Management Strategies in Women with Endometriosis Undergoing IVF

### 4.1. Pre-Treatment

The use of prolonged GnRH analogues in women with endometriosis prior to ovarian stimulation has been proposed to improve pregnancy outcomes. The continuous exposure of the hypothalamic–pituitary ovarian axis to GnRHa causes the downregulation of GnRH receptors and desensitization of the pituitary gland, resulting in a period of prolonged amenorrhoea and low estradiol levels, which is proposed to mitigate the inflammatory effects associated with endometriosis [45]. A recent Cochrane review concluded that the benefit of prolonged GnRH therapy (minimum 3 months) versus no pre-treatment prior to IVF or ICSI is uncertain with regards to clinical pregnancy, miscarriage, and live birth [46]. Concerns with prolonged GnRH analogue use include pituitary over-suppression resulting in poor ovarian response and luteal insufficiency [47], although the effect of prolonged GnRH analogue use on the mean oocyte and embryo numbers is uncertain based on existing low-quality data, and further studies are needed [46].

The post-operative use of GnRH analogue therapy for endometriosis has been suggested to reduce the risk of disease recurrence and endometrioma formation with potential for use in the management of symptomatic women who are awaiting IVF [48]. However, a randomized control trial on 400 women with minimal and mild endometriosis undergoing IVF who were treated with prolonged GnRH analogues for 3 months following surgical cauterization of endometriosis before undergoing IVF treatment showed no benefit with regards to embryo quality, implantation rates, or clinical pregnancy rates [49]. Therefore, extended downregulation in women with endometriosis prior to IVF either following surgery or without surgery does not seem to confer benefit in improving IVF outcomes.

The use of Letrozole, an aromatase inhibitor, has been evaluated in one retrospective cohort study including 126 women with endometriomas, showing improved reproductive outcomes when used along with GnRH analogues [50]. Further studies evaluating the use of Letrozole in women with endometriosis undergoing IVF are needed.

The use of long-term dienogest therapy in women with endometriosis (*n* = 723) undergoing IVF treatment has recently been evaluated in a systematic review and meta-analysis by Reiter and colleagues, showing no significant difference in clinical pregnancy rate, live birth rate, miscarriage rate, or number of oocytes retrieved [51].

An ongoing randomized double-blinded trial comparing the effectiveness of two weeks of Elagolix, an oral GnRH receptor antagonist versus placebo prior to IVF in patients with endometriosis, will shed light on the use of this class of drugs that have recently been shown to be effective for the improvement of endometriosis-related symptoms [52,53].

Results from a non-inferiority randomized controlled trial evaluating 3 month pre-treatment with either the combined oral contraceptive pill or GnRH analogue in women with surgically diagnosed stage III–IV endometriosis are currently awaited [54].

### 4.2. Ovarian Stimulation

There remains a lack of studies evaluating gonadotropin preparations or dosages for performing ovarian stimulation in women with endometriosis [55]. Reassuringly, performing IVF does not increase disease progression or recurrence in deep endometriosis patients [56,57].

There is low-quality evidence that women with endometriosis may require a higher dose of gonadotrophins as they have a lower response to gonadotropins during ovarian stimulation, with one study including 40 women with ovarian endometriosis, and 80 women with tubal infertility undergoing IVF treatment showing a lower ovarian response and significantly higher dose of urinary FSH used, although there was no difference in live birth rates between groups [55,58]. Further research in this area is needed.

It appears that there is no specific approach to ovarian stimulation in women with endometriosis, although a higher dose of gonadotrophins may be required in women with endometriosis. Other factors may need to be taken into consideration, including the approach of the clinical team and patient preference.

### 4.3. GnRH Agonist vs. Antagonist for Downregulation during Ovarian Stimulation

A randomized controlled trial in women with minimal and mild endometriosis and endometrioma who underwent 246 cycles of IVF with either the GnRH agonist or the GnRH antagonist protocols reported equivalent implantation and clinical pregnancy rates in both protocols [59]. A systematic review and meta-analysis in women with moderate and severe endometriosis undergoing IVF with either the short or long GnRH agonist protocol showed higher clinical pregnancy rates in women who received the long GnRH agonist protocol; however, in their subgroup analysis of 14 non-randomized controlled trial studies of GnRH agonist downregulation protocols by duration (short, long and ultralong), there was no significant difference in reproductive outcomes [60].

Current evidence suggests no difference in reproductive outcomes between GnRH agonist and GnRH antagonist downregulation protocols; however, if GnRH agonist downregulation is used, a longer protocol could be considered [61], particularly following surgical management for symptomatic endometriosis.

### 4.4. Pelvic Infection

The risk of pelvic infection and abscess formation following oocyte retrieval may be increased in the presence of an ovarian endometrioma, although the overall risk remains low, reported at 0.12% according to a retrospective study on 5958 transvaginal ultrasound-scan-guided oocyte retrieval procedures [62]. It is important to counsel women with endometriosis undergoing oocyte retrieval regarding the need to report symptoms suggestive of infection. The use of an aseptic technique for preparation and the routine use of broad-spectrum infection seems to reduce infection rates to a minimal level [63]. The contamination of retrieved oocytes with endometrioma fluid may be avoided through the careful positioning of the aspiration needle at oocyte retrieval. If endometrioma puncture occurs, it is recommended to consider an extended course of antibiotics.

### 4.5. Method of Fertilization

There remains a lack of data regarding the impact of IVF and ICSI on fertilization rates and pregnancy rates in women with endometriosis. A retrospective study on 503 IVF cycles reassuringly found no significant difference in fertilization rates using IVF in women with endometriosis compared to women with infertility due to other causes [30]. Further research is needed to evaluate whether there is any benefit in using ICSI in women with endometriosis in the absence of male factor infertility.

### 4.6. Elective Embryo Freezing

The supraphysiological rise in estradiol levels during ovarian stimulation is proposed to exacerbate endometriosis, adversely affect endometrial receptivity, and impair implantation [64]. Elective embryo freezing and deferred frozen embryo transfer has been suggested as an approach to IVF treatment in women with endometriosis to mitigate these adverse effects. In a retrospective cohort study on women with endometriosis (*n* = 135) undergoing either fresh or deferred frozen embryo transfer, there were higher cumulative pregnancy rates in the frozen embryo transfer group (34.8% vs. 17.8%, *p* = 0.0005), although there was no significant difference in the live birth rate [64]. Conversely, a meta-analysis of 3010 women with endometriosis showed higher live birth rates following frozen embryo transfer compared to the fresh-embryo transfer group (OR 1.53, 95% CI 1.13–2.08, *p* = 0.007) [65]. Further studies are needed to establish the true benefit of deferred frozen embryo transfer, which remains uncertain based on current evidence.

### 4.7. Surgical Treatment of Endometriosis

There are several studies investigating the use of surgical treatment in women with endometriosis undergoing IVF treatment, but there remains a need for randomized controlled trials.

Several systematic reviews and meta-analyses have concluded that ovarian cystectomy for endometrioma in women undergoing IVF does not improve clinical pregnancy or live birth rates [14,16,66]. In addition, ovarian cystectomy prior to IVF treatment can lead to a higher rates of cycle cancellation due to poor ovarian response and failed oocyte retrieval with no significant difference in live birth rates compared to women who had no surgical treatment [67]. Further systematic reviews and meta-analyses looked at the possible benefit of surgery for endometriomas and similarly reported no significant difference in pregnancy rates compared to other treatment regimens, including surgery together with IVF, IVF only, and the aspiration of endometriomas with subsequent IVF [68,69].

Outcomes in patients who underwent surgery for deep infiltrative endometriosis (DIE) before IVF were compared with those in patients who underwent IVF without previous surgery for DIE in a meta-analysis by Casals et al. in 2021 [70]. Notwithstanding concerns regarding the accuracy of their analysis and the significant loss to follow up, the authors concluded that surgery prior to IVF treatment in patients with DIE may offer benefit [70]. Notably, surgery for DIE can be complex, carrying an overall 18.5% risk of post-operative complications, and therefore, it needs to be considered very carefully in symptomatic patients [57,71].

Opoien et al. included 661 women in their study comparing the effectiveness of surgery for the treatment of minimal and mild endometriosis with diagnostic laparoscopy alone prior to embarking on IVF treatment [72]. They reported higher implantation, pregnancy, and live birth rates in women who underwent surgical removal of all visible ASRM stage I and II endometriosis lesions compared with controls who received only diagnostic laparoscopies. The surgical treatment group was found to have a shorter waiting time to achieve a pregnancy as well as higher rates of cumulative pregnancy. However, it is worth noting that the benefits appear minimal, with the number needed to treat being 14 [72]. Importantly, there was a substantial difference in the duration of infertility and the time interval between surgery and IVF treatment, which may solely explain the slight difference in live birth rates (20.6% vs. 27.7%) [72].

A recent systematic review and meta-analysis by Bourdon and colleagues reported that surgery prior to IVF does not improve IVF/ICSI outcomes [73]. Moreover, subgroup analysis of higher quality studies suggests that IVF/ICSI outcomes may be lower in women with a history of previous surgery [73].

Overall, surgery for ovarian endometrioma prior to surgery may be considered for endometriosis-associated symptom control or to improve follicular accessibility for oocyte retrieval; however, it should not be performed to improve IVF outcomes or prevent disease progression, as there currently is a lack of data to support this approach. There remains a need for further studies to evaluate the benefits of performing surgery for endometriosis prior to IVF treatment with the aim of improving IVF outcomes.

## 5. Conclusions

The presence of endometriosis may adversely impact ovarian reserve, oocyte/embryo quality, response to ovarian stimulation, and accessibility for oocyte retrieval during IVF treatment. The analysis of data from large registries for assisted reproduction technology and several meta-analyses has shown that women with endometriosis have lower oocytes retrieved but maintain similar livebirth rates to those of women without endometriosis. Surgery for ovarian endometriomas prior to IVF treatment should not be performed routinely, since this is detrimental to ovarian reserve and does not improve live birth rates; however, careful consideration should be applied to its use in the context of endometriosis symptom control and to improve access to ovarian follicles for egg collection. Reassuringly, performing IVF does not increase disease progression or recurrence in deep endometriosis patients. Future studies in the form of well-designed randomized controlled trials are needed to further evaluate the role of surgical and medical treatment options in women with endometriosis undergoing assisted conception.

## Figures and Tables

**Table 1 medicina-60-01358-t001:** A list of the studies that have examined the effect of endometriosis on oocyte and embryo quality and an overview of their findings, Reproduced with permission from the *Journal of Clinical Medicine*, “Special issue on Endometriosis and infertility: Insights into causal links management strategies”, S Latif and E Saridogan 2023 [23].

Scheme	Study Design	Impact of Endometriosis on Oocyte/Embryo Quality	Study Findings
Goud et al., 2014 [24]	Prospective cohort study, *n* = 28 women	Oocyte quality reduced ↓	- Increased likelihood of oocyte to fail in vitro maturation IVM; - Altered oocyte morphology (cortical granule loss, spindle disruption, zona pellucida hardening).
Kasopoglu et al., 2017 [25]	Retrospective cohort study, *n* = 72 women	Oocyte quality reduced ↓	- Altered oocyte morphology (morphological abnormalities of the cytoplasm, zona pellucida, and first polar body).
Simon et al., 1994; Sung et al., 1997; Diaz et al., 2000 [4,26,27]	Restrospective cohort study, *n* = 137 women; retrospective cohort study, *n* = 239 women; matched case–control study, *n* = 58 women	Oocyte quality reduced ↓	- Lower implantation rates in donor oocytes from women with endometriosis; - Equivalent implantation rates and pregnancy rates when women with endometriosis using donor oocytes from healthy women.
Ferrero et al., 2019 [28]	Prospective cohort, *n* = 12 women	Oocyte quality reduced ↓	- Differential transcriptomic profile associated with lower oocyte quality.
Sanchez et al., 2017 [9]	Review article	Oocyte quality reduced ↓	- Altered oocyte morphology; - Increased likelihood of oocyte to fail IVM; - Lower cytoplasmic mitochondrial content.
Robin et al., 2021 [29]	Retrospective cohort study, *n* = 596 women	Oocyte quality unaffected ↔	- Normal oocyte morphology; - Lower number of top-quality embryos and lower cumulative clinical pregnancy rate are both attributed to lower oocyte yield.
Metzemaeker et al., 2020, Filippi et al., 2014, Yang et al., 2015, Hamdan et al., 2015 [14,16,30,31]	Population-based cohort study, *n* = 503 IVF cycles [30]; prospective cohort study, *n* = 29 women [31]; systematic review and meta-analysis, *n* = 1039 women [16]; systematic review and meta-analysis, *n* = 928 women [14]	Oocyte quality unaffected ↔	- Equivalent fertilization rates.
Brizek et al., 1995 [32]	Retrospective cohort study, *n* = 235 embryos	Embryo quality reduced ↓	- Increased incidence of aberrant development of embryos (more prevalent nuclear and cytoplasmic impairment, cytoplasmic fragmentation, uneven cleavage).
Pellicer et al., 2001 [5]	Retrospective cohort study, n= 70 women	Embryo quality reduced ↓	- Altered embryo morphology (fewer blastomeres per embryo, a higher number of arrested embryos.
Paffoni et al., 2019 [33]	Randomized-controlled in vitro study, *n* = 147 oocytes	Embryo quality reduced ↓	- Altered embryo morphology (excess cellular fragmentation, cell death in blastomeres, and altered blastomere division.
Alshehre et al., 2020 [17]	Systematic review and meta-analysis, *n* = 8 studies	Embryo quality unaffected ↔	- No difference in total number of embryos; - No difference in number of top-quality embryos; - No difference in clinical pregnancy rate, implantation rate, or live birth rate.
Sanchez et al., 2020 [34]	Retrospective matched cohort study, *n* = 3818 embryos	Embryo quality unaffected ↔	- Equivalent number of cleavage embryos; - Equivalent number of good-quality embryos.
Dongye et al., 2021 [35]	Systematic review and meta-analysis, *n* = 22 studies	Embryo quality unaffected ↔	- Normal embryo morphology.
Juneau et al., 2017 [21]	Retrospective cohort study, *n* = 305	Embryo quality unaffected ↔	- No difference in aneuploidy rates.

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
