# Peer review of "Endometriosis and In Vitro Fertilization"

_medicina, 2024, doi:10.3390/medicina60081358_

Round 1

Reviewer 1 Report

Comments and Suggestions for Authors

These lines need corrections.
Restrictions to the English language and human species.
We 34 screened references of the selected articles to identify other studies of relevance.

In addition, both oocyte 40 yield and fertilization rate were affected by endometriosis of any stage.

Overall, although presence of endometriosis is considered to impact oocyte quality and in return embryo quality, a clear clinical impact on IVF outcomes is not confirmed.

The eutopic endometrium of women with endometriosis has been to shown to exhibit characteristics of progesterone resistance and aberrant cell signaling, suggested to alter endometrial receptivity [39].

Comments on the Quality of English Language

Author Response

  1. Reviewer 1’s comment
  • “ Summary: The main research question was to what extent does endometriosis impact the success of in vitro fertilization (IVF) treatment, particularly in terms of oocyte and embryo quality and endometrial receptivity?

It was concluded that Endometriosis negatively affects ovarian reserve and oocyte/embryo quality but does not reduce live birth rates in IVF. Routine surgery for ovarian endometriomas is not recommended due to its adverse effects on ovarian reserve. Further randomized controlled trials are needed to assess treatment options for women with endometriosis undergoing assisted conception.

It is a very well written review considering the importance of the topic questions addressed and the large number of articles they have gone through to reach the conclusion.”

  1. Author’s response to Reviewer 1’s comment

Many thanks for this comment.

  1. Reviewer 1’s comment

“Minor Issues: Some grammatical errors were found.

1.Restrictions to the English language and human species. We screened references of the selected articles to identify other studies of relevance.

2.In addition, both oocyte yield and fertilization rate were affected by endometriosis of any stage.

3.Overall, although presence of endometriosis is considered to impact oocyte quality and in return embryo quality, a clear clinical impact on IVF outcomes is not confirmed.

4.The eutopic endometrium of women with endometriosis has been to shown to exhibit characteristics of progesterone resistance and aberrant cell signaling, suggested to alter endometrial receptivity [39].”

  1. Author’s response to Reviewer 1’s comment

Thank you for pointing out these grammatical errors. All four grammatical errors have been amended in the manuscript on line 40 page 1, line 51 page 2, line 82 page 2 and line 91 page 3.

  1. Reviewer 1’s comment

“Recommendation: I recommend this review owing to the strenuous efforts and intensive literature search by the authors and reaching a solid conclusion. It will definitely contribute to the existing literature and help IVF experts.”

  1. Author’s response to Reviewer 1’s comment

Many thanks for recommending our review.

Reviewer 2 Report

Comments and Suggestions for Authors

In this study authors gathered some information about endometriosis and IVF, in a beautiful shape, but few original information added. I think this work can hardly qualify as a narrative review.

-First of all, in Methods, authors wrote: “ A PUBMED search was performed for articles published between October 1991 and May 2024…. Restrictions to the English language and human species. We screened references of the selected articles to identify other studies of relevance.” Is this all? That means that you analyzed altogether clinical trials and meta-analyses and case reports and letters and anything? Really?

-I think that you could have made a Flow diagram to show that you found x studies, you chose only clinical trials, and eliminated the rest, there were y clinical trials dealing with endometriosis and IVF, that you finally studied. OR that there were few studies about this topic, therefore you chose to study all that was written after 2019 (recent titles, with advanced technique).

-Then, in each chapter that you presented, readers have no clue how many studies you studied about that specific topic, what type of studies, or there are very few studies about that topic. For example, you wrote:

3.2. Endometrial receptivity  

Endometrial receptivity is dependent upon appropriate hormonal exposure of oestrogen and progesterone and the regulated expression of numerous genes [39]. The eutopic endometrium of women with endometriosis has been to shown to exhibit characteristics of progesterone resistance and aberrant cell signaling, suggested to alter endometrial receptivity [39]. However, in their large retrospective cohort study including 459 euploid frozen embryo transfer cycles comparing women with surgically-diagnosed endometriosis to women undergoing IVF for isolated male factor infertility, Bishop et al., found that there was no difference in implantation rate, clinical pregnancy, pregnancy loss, or live- birth rate [21]. They reported that blastocyst development, number of good-quality blastocysts produced and euploidy rates were the same in both groups. This suggests that when known high-quality embryos are transferred during a medicated frozen embryo transfer cycle, women with endometriosis do not exhibit any marked defect in endome-87 trial receptivity [21].”

You came to a conclusion using only two studies (39 and 21).  The second one was a large retrospective cohort study, while the first was a review.

-Most of the Reference titles are written long before 2019, and only 24 out of 68 are written after 2019.

Author Response

Reviewer 2

Reviewer 2’s comment

“In this study authors gathered some information about endometriosis and IVF, in a beautiful shape, but few original information added. I think this work can hardly qualify as a narrative review.

Author’s response to Reviewer 2’s comment

Thank you for this comment. We have now undertaken major revisions to improve the manuscript as detailed in our responses to your subsequent comments.

Reviewer 2’s comment

-First of all, in Methods, authors wrote: “ A PUBMED search was performed for articles published between October 1991 and May 2024…. Restrictions to the English language and human species. We screened references of the selected articles to identify other studies of relevance.” Is this all? That means that you analyzed altogether clinical trials and meta-analyses and case reports and letters and anything? Really?”

Author’s response to Reviewer 2’s comment

Thank you for this comment. We have now updated the manuscript on line 42, page 2 to clarify that for this review we included clinical trials, systematic reviews, meta-analyses and clinical studies deemed to be of relevance.

Reviewer 2’s comment

“-I think that you could have made a Flow diagram to show that you found x studies, you chose only clinical trials, and eliminated the rest, there were y clinical trials dealing with endometriosis and IVF, that you finally studied. OR that there were few studies about this topic, therefore you chose to study all that was written after 2019 (recent titles, with advanced technique).”

Author’s response to Reviewer 2’s comment

Many thanks for this comment. As advised, we have now updated the manuscript to highlight the paucity of data in certain areas of endometriosis research (on line 149 page 4, line 165 page 4, line 171 page 4, line 176 page 4, line 182 page 4, line 191 page 5, line 229 page 6, line 243 page 6 and line 287 page 7) with a view to highlighting knowledge gaps where there is a need for further studies. There are several important clinical studies prior to 2019 and therefore these remain included.

Reviewer 2’s comment

“-Then, in each chapter that you presented, readers have no clue how many studies you studied about that specific topic, what type of studies, or there are very few studies about that topic. For example, you wrote:

“3.2. Endometrial receptivity  

Endometrial receptivity is dependent upon appropriate hormonal exposure of oestrogen and progesterone and the regulated expression of numerous genes [39]. The eutopic endometrium of women with endometriosis has been to shown to exhibit characteristics of progesterone resistance and aberrant cell signaling, suggested to alter endometrial receptivity [39]. However, in their large retrospective cohort study including 459 euploid frozen embryo transfer cycles comparing women with surgically-diagnosed endometriosis to women undergoing IVF for isolated male factor infertility, Bishop et al., found that there was no difference in implantation rate, clinical pregnancy, pregnancy loss, or live- birth rate [21]. They reported that blastocyst development, number of good-quality blastocysts produced and euploidy rates were the same in both groups. This suggests that when known high-quality embryos are transferred during a medicated frozen embryo transfer cycle, women with endometriosis do not exhibit any marked defect in endometrial receptivity [21].”

You came to a conclusion using only two studies (39 and 21).  The second one was a large retrospective cohort study, while the first was a review.”

Author’s response to Reviewer 2’s comment

Thank you for this comment, as advised we have now expanded the manuscript for the section on endometrial receptivity on line 88 page 3 and highlighted throughout the manuscript where limited data is currently available (on line 149 page 4, line 165 page 4, line 171 page 4, line 176 page 4, line 182 page 4, line 191 page 5, line 229 page 6, line 243 page 6 and line 287 page 7).

Reviewer 2’s comment

“-Most of the Reference titles are written long before 2019, and only 24 out of 68 are written after 2019.”

Author’s response to Reviewer 2’s comment

Thank you for this comment, as advised we have now updated the manuscript to include the below recent studies. Earlier important clinical studies remain included.

  • Zondervan KT, Becker CM, Missmer SA. Endometriosis. New Engl J Med. 2020;382(13):1244–56.
  • Hodgson RM, Lee HL, Wang R, Mol BW, Johnson N. Interventions for endometriosis-related infertility: a systematic review and network meta-analysis. Fertil Steril. 2020 Feb;113(2):374-382.e2. doi: 10.1016/j.fertnstert.2019.09.031.
  • Lensen S, Osavlyuk D, Armstrong S, Stadelmann C, Hennes A, Napier E, Wilkinson J, Sadler L, Gupta D, Strandell A, Bergh C, Vigneswaran K, Teh WT, Hamoda H, Webber L, Wakeman SA, Searle L, Bhide P, McDowell S, Peeraer K, Khalaf Y, Farquhar C. A Randomized Trial of Endometrial Scratching before In Vitro Fertilization. N Engl J Med. 2019;380(4):325–34.
  • van der Houwen LEE, Lier MCI, Schreurs AMF, van Wely M, Hompes PGA, Cantineau AEP, Schats R, Lambalk CB, Mijatovic V. Continuous oral contraceptives versus long-term pituitary desensitization prior to IVF/ICSI in moderate to severe endometriosis: study protocol of a non-inferiority randomized controlled trial. Hum Reprod Open. 2019 Feb 23;2019(1):hoz001. doi: 10.1093/hropen/hoz001
  • Reiter A, Balayla J, Dahdouh EM, Awwad JT. The Effects of Long-Term Dienogest Therapy on In Vitro Fertilization Outcomes in Women with Endometriosis: A Systematic Review and Meta-Analysis. J Obstet Gynaecol Can. 2024 Apr;46(4):102339. doi: 10.1016/j.jogc.2023.102339. 
  • ESHRE Guideline Group on Ovarian Stimulation, Bosch E, Broer S, Griesinger G, Grynberg M, Humaidan P, Kolibianakis E, Kunicki M, Marca AL, Lainas G, Clef NL, Massin N, Mastenbroek S, Polyzos N, Sunkara SK, Timeva T, Töyli M, Urbancsek J, Vermeulen N, Broekmans F. ESHRE guideline: ovarian stimulation for IVF/ICSI†. Hum Reprod Open. 2020;2020(2):hoaa009.
  • Somigliana E, Viganò P, Benaglia L, Busnelli A, Paffoni A, Vercellini P. Ovarian stimulation and endometriosis progression or recurrence: a systematic review. Reprod Biomed Online. 2019;38(2):185–94.
  • Mijatovic V, Vercellini P. Towards comprehensive management of symptomatic endometriosis: beyond the dichotomy of medical versus surgical treatment. Hum Reprod. 2024;39(3):464–77.
  • Balla A, Quaresima S, Subiela JD, Shalaby M, Petrella G, Sileri P. Outcomes after rectosigmoid resection for endometriosis: a systematic literature review. Int J Colorectal Dis 2018;33:835–847.

Round 2

Reviewer 2 Report

Comments and Suggestions for Authors

Thank you for the changes you made!